# Direct Interaction of Coronavirus Nonstructural Protein 3 with Melanoma Differentiation-Associated Gene 5 Modulates Type I Interferon Response during Coronavirus Infection

**DOI:** 10.3390/ijms231911692

**Published:** 2022-10-02

**Authors:** Xinxin Sun, Li Quan, Ruiai Chen, Dingxiang Liu

**Affiliations:** 1Zhaoqing Branch Center of Guangdong Laboratory for Lingnan Modern Agricultural Science and Technology, Zhaoqing 526000, China; 2Integrative Microbiology Research Centre, South China Agricultural University, Guangzhou 510642, China

**Keywords:** coronavirus, SARS-CoV-2, IBV, nsp3, MDA5, type I IFN response

## Abstract

Coronavirus nonstructural protein 3 (nsp3) is a multi-functional protein, playing a critical role in viral replication and in regulating host antiviral innate immunity. In this study, we demonstrate that nsp3 from severe acute respiratory syndrome coronavirus 2 (SARS-CoV-2) and avian coronavirus infectious bronchitis virus (IBV) directly interacts with melanoma differentiation-associated gene 5 (MDA5), rendering an inhibitory effect on the MDA5-mediated type I interferon (IFN) response. By the co-expression of MDA5 with wild-type and truncated nsp3 constructs, at least three interacting regions mapped to the papain-like protease (PLpro) domain and two other domains located at the N- and C-terminal regions were identified in SARS-CoV-2 nsp3. Furthermore, by introducing point mutations to the catalytic triad, the deubiquitylation activity of the PLpro domain from both SARS-CoV-2 and IBV nsp3 was shown to be responsible for the suppression of the MDA5-mediated type I IFN response. It was also demonstrated that both MDA5 and nsp3 were able to interact with ubiquitin and ubiquitinated proteins, contributing to the interaction between the two proteins. This study confirms the antagonistic role of nsp3 in the MDA5-mediated type I IFN signaling, highlighting the complex interaction between a multi-functional viral protein and the innate immune response.

## 1. Introduction

Coronavirus (CoV) is a main human and animal pathogen. The emergence of severe acute respiratory syndrome coronavirus (SARS-CoV) originated from bats in 2003, Middle East respiratory syndrome coronavirus (MERS-CoV) in 2012, and the current COVID-19 pandemic caused by severe acute respiratory syndrome coronavirus 2 (SARS-CoV-2) calls for urgent development of effective intervention measures [1,2]. Understanding the molecular functions of individual CoV proteins related to viral replication and virus–host interactions, especially with host antiviral mechanisms, would be essential for the development of such tools.

Coronavirus encodes a single-stranded, positive-sense RNA genome of 27–32 kb [3]. Two-thirds of the virus genome translates into two large polyproteins, pp1a and pp1ab, which are cleaved by viral proteases into 15–16 non-structural proteins (nsp1–16). The papain-like protease (PLpro) encoded by nsp3 and 3CLpro encoded by nsp5 are two virus-encoded proteases, cleaving the two polyproteins [4,5,6]. The mature nsps are assembled to form a multi-functional membrane-bound replicase complex (RTC), mediating the replication of the viral genome and initiating the transcription and translation [7]. One critical component is nsp3, a multi-functional protein containing a number of putative domains. So far, eight putative domains, including Ubl (ubiquitin-like domain1), the hypervariable region, X-domain (also called macrodomain), ubl2 (ubiquitin-like domain 2), the papain-like protease 2 (PLpro or PL2^pro^), the nsp3 ectodomain, Y and Y1 domains, are identified in almost all coronaviruses [8]. Except for the PLpro domain, the functionality of other domains is poorly characterized. The presence of these multi-functional domains in nsp3 makes it an attractive therapeutic target.

As the first line of host defense against pathogen invasion, the innate immune system can detect pathogen-associated molecular patterns (PAMPs) through pattern recognition receptors (PRR) and initiate a series of signal cascades to induce the expression of antiviral genes [9]. The antiviral response to cytosolic viral RNA is mediated by retinoic acid-inducible gene I (RIG-I)-like receptors (RLR). Melanoma differentiation-associated gene 5 (MDA5), a member of the RLRs family, consists of the N terminal tandem highly conservative caspase recruitment domains (CARDS), a central DExD/H box RNA helicase domain with an ATP binding motif, and a C-terminal domain (CTD) [10,11]. When MDA5 recognizes double-strand RNAs, the viral replication intermediates, protein phosphatase 1 (pp1) α/γ dephosphorylates the CARDs [12]. This promotes the binding and activation of mitochondrial antiviral-signaling protein (MAVS), leading to the phosphorylation of IRF3/IRF7 and the induction of type I interferons [13,14].

Avian coronavirus infectious bronchitis virus (IBV) infects chickens and causes respiratory symptoms and kidney and oviduct diseases [15]. Due to the absence of RIG-I in chicken [16], MDA5 would play a major role in recognizing IBV as well as other coronavirus RNA. In this study, we show that the full-length nsp3 from SARS-CoV-2 and IBV directly interacts with MDA5, forming complexes with MDA5, ubiquitin (Ub), and ubiquitinated proteins. The further functional characterization of these complex interactions demonstrated that the ubiquitylation of MDA5 enhances the MDA5-mediated IFN-β induction, whereas the deubiquitinating (DUB) activity of nsp3 PLpro demolishes this enhancement effect. 

## 2. Material and Method

### 2.1. Antibodies, Reagents, Cell Culture and Virus Infection

The antibodies against β-actin, Flag, and HA were purchased from TansGen biotech (Beijing, China). The antibodies against MDA5 were purchased from proteintech (Wuhan, China). Antiserum against IBV N protein was prepared from rabbits immunized with bacterially expressed fusion proteins.

The H1299 and HEK293T cells were cultured in complete Dulbecco’s modified Eagle’s medium (DMEM) (Gibco, Shanghai, China) supplemented with 8% fetal bovine serum (FBS), penicillin (100 units/mL), and streptomycin (100 μg/mL). All of the cells were cultured at 37 °C with 5% CO_2_.

Vero-adapted IBV (p65) was obtained from the American Type Culture Collection (ATCC) and passaged in Vero cells 65 times, as previously described [17,18]. The rIBV-HA-3541 was rescued using the reverse genetic technique. The virus stocks were prepared by infecting monolayers of Vero cells with rIBVs at a multiplicity of infection (MOI) of 0.1, and the cell lysates were harvested when the complete fusion of the entire monolayer was observed. After repeated freezing and thawing three times, the total cell lysates were clarified by centrifugation at 5000× *g* at 4 °C for 10 min and stored at −80 °C.

In the infection experiments, the cells were washed with PBS before being infected with rIBVs and incubated in serum-free DMEM at an MOI of approximately 2. After 2 h of absorption, the cells were washed twice with PBS to remove the unbound viruses.

### 2.2. RNA Extraction and RT-PCR Analysis

The total RNA was extracted from the cells using TRIzol reagent (TansGen biotech, Beijing, China), and the first-strand cDNA was synthesized by reverse transcriptase M-MLV (Takara, Beijing, China) with specific primers. The relative transcriptional level of IFN-β was detected by fluorescent quantitative real-time PCR (qPCR), using specific primers based on the human IFN-β sequence.

The above cDNA (1 μL of the 20 μL RT reaction mixture) was used as templates and subjected to SYBR green PCR assays (TaKaRa, Beijing, China) at least three times. qPCR was performed using the CFX96 Touch Real-Time PCR Detection System (Bio-Rad, Hercules, CA, USA), as follows: initial denaturation at 95 °C for 3 min followed by 40 cycles at 95 °C for 15 s, 56 °C. for 30 s, and 72 °C for 30 s. A final melting curve analysis was performed from 65 °C to 95 °C at a rate of 0.1 °C/s (continuous acquisition). The results are expressed as the relative gene expression level with normalization to the expression level of glyceraldehyde-3-phosphate dehydrogenase (GAPDH). The relative mRNA level was calculated using the 2^−ΔΔCt^ method.

### 2.3. Plasmid Construction and Transfection

Human MDA5 was amplified from the total RNA of 293T cells by RT-PCR, using primer pair: TAGGGCGAATTCGGATCCatgtcgaatgggtattccacag and CGCCTCGAGAAGCTTctaatcctcatcactaaataaacagcattc. The PCR product was inserted into vector pXJ40-Flag by homologous recombination. The SARS-CoV-2 nsp3 cDNA was synthesized (GENEWIZ) based on the SARS-CoV-2 nsp3 sequence (NCBI MT827843) and inserted into plasmid K1E-Hibit. The IBV nsp3 cDNA was reverse-transcribed and amplified from the total RNA extracted from the IBV-infected H1299 cells, and cloned into pXJ40 and pK1E vectors, respectively, by homologous recombination. All of the plasmids were verified by sequencing.

The HEK293T cells were plated to a 12- or 6-well plate and transfected with plasmid DNA when the cell density was about 70%, using the TransIntro EL reagent (Transgen Biotech, Beijing, China). According to the manufacturer’s instruction, 1.6 μg plasmid DNA and 3.2 μL TransIntro EL were diluted with 100 μL Opti-MEM (Gibco) and incubated for 20 min. The cells were washed with PBS before being transfected and incubated in 900 μL Opti-MEM containing 10% FBS.

### 2.4. Western Blot Analysis

The cells were collected with a cell scraper (Corning) at a specified time point post-transfection. After centrifugation for 1 min at 16,000× *g*, the cell pellets were dissolved in 1 × RIPA buffer. After clarification, the protein concentration was determined using the BCA protein Assay Kit (Beyotime, Shanghai, China), according to the manufacturer’s instructions. The cell lysates were mixed with Laemmli’s sample buffer containing 100 mM dithiothreitol, boiled at 90 °C for 5 min, and centrifuged at 16,000× *g* for 5 min. Equal amounts of protein samples were subjected to sodium dodecyl sulphate–polyacrylamide gel electrophoresis and transferred to 0.2 µM BIORAD nitrocellulose membrane (BIORAD). After blocking the non-specific antibody binding with 5% skim milk in tris-buffer saline containing 0.1% Tween 20 (20 mM Tris-HCl pH 7.4, 150 mM NaCl), the membrane was incubated overnight with 1 μg/mL primary antibody at 4 °C. After washing, 1:2000 diluted anti-rat or anti-rabbit IgG antibody coupled with horseradish peroxidase (DAKO) was added and incubated at room temperature for 2 h. The proteins were detected using the chemiluminescence detection kit (Amersham Biosciences, Amersham, UK) and medical X-ray film (Fuji film, Tokyo, Japan) according to the manufacturer’s instructions.

### 2.5. Co-Immunoprecipitation

The cells transfected with appropriate plasmid DNA or infected with the virus were lysed with 300 μL of RIPA buffer per well (6 well plate), 3 μL of protease inhibitor was added, mixed well, and placed on a table rotator at 4 °C for 15 min. After centrifugation at 10,600× *g* for 10 min, the supernatant was transferred to a fresh tube containing washed beads and added lysis buffer to keep the total volume to 1 mL, 10 μL protease inhibitors were added and incubated at 4 °C for 2 h. After centrifugation at 8300× *g*, 4 °C for 40 s, the supernatant was discarded, and the pellets were washed with 1 mL lysis buffer three times. The principates were then analyzed by Western blot.

### 2.6. Immunofluorescence

The H1299 cells cultured in 48-well plates to confluency were infected with rIBV-HA-3541 at MOI of ~2. At 20 h post-infection, the cells were fixed with ice-cold 100% methanol for 15 min at −20 °C, rinsed 3 times with 1 × PBS for 5 min each. The cells were then incubated with blocking buffer at room temperature for 60 min, diluted primary antibody was added and incubated at 4 °C overnight, and rinsed 3 times with 1 × PBS for 5 min each. After incubation with diluted secondary antibody for 1–2 h at room temperature in the dark, the cell nuclei were stained by adding 10 µg/mL Hoechst 33342 and incubated at room temperature for 5–10 min, rinsed 3 times with 1 × PBS for 5 min each. Finally, the coverslips were mounted using the antifade reagent, and the cells were examined immediately or stored at 4 °C in the dark.

## 3. Result

### 3.1. Efficient Expression of the Full-Length SARS-CoV-2 and IBV nsp3

Coronavirus nsp3 is a multi-domain protein, but the functional characterization of the protein was hampered by the lack of an expression system that can efficiently express the full-length nsp3 protein [19]. In this study, an efficient two-plasmid expression system was first developed based on the K1E phage RNA polymerase and a capping enzyme from the African swine fever virus [20]. The co-transfection of the two plasmids together with target genes cloned under the control of the KIE promoter would result in the cytoplasmic expression of the target genes. As shown in Figure 1, the full-length SARS-CoV-2 nsp3 with either an HA or a Hibit tag at the N-terminus (219.3 kDa) and the full-length IBV nsp3 with a Hibit or a FLAG tag (180.5 kDa) were highly efficiently expressed. This expression system was subsequently used to study the interactions between coronavirus nsp3 and host proteins.

The HEK293T cells were transfected with pK1E (empty vector), pK1E-Hibit-IBV, SARS2-nsp3, pK1E-FLAG-IBV, and pK1E-HA-SARS2-nsp3, respectively. The cells were harvested at 24 h post-transfection and subjected to Western blot with indicated antibodies. Beta-actin was included as the loading control. Numbers on the left indicate protein sizes in kilodalton.

### 3.2. Direct Interaction of SARS-CoV-2 and IBV nsp3 with MDA5

As MDA5 plays an essential role in anti-coronavirus innate immunity and was previously shown to interact with SARS-CoV-2 nsp3 in virus-infected cells [21], it would be interesting to investigate if the full-length SARS-CoV-2 nsp3 directly interacts with MDA5 in the absence of other viral proteins and viral RNA, and regulates the functions of MDA5. The direct interaction of SARS-CoV-2 nsp3 with MDA5 was first analyzed by co-expression of the two proteins as Hibit-tagged nsp3 (Hibit-SARS2-nsp3) and Flag-tagged MDA5 (Flag-MDA5) in 293T cells. The cells transfected with an empty vector plus one target protein (Hibit-SARS2-nsp3+Flag and Hibit+Flag-MDA5) were included as controls. Following immunoprecipitation with anti-Flag coated beads, the co-precipitated proteins were analyzed by Western blot using anti-Flag and anti-Hibit antibodies, respectively. As shown in Figure 2a, similar amounts of Hibit-SARS2-nsp3 were detected in cells either transfected with Hibit-SARS2-nsp3+Flag or co-transfected with Flag-MDA5 (Hibit-SARS2-nsp3+Flag-MDA5). In the immunoprecipitation assay, Hibit-SARS2-nsp3 was only detected when it was co-expressed with Flag-MDA5 (Figure 2a). These results confirm the direct interaction between SARS-CoV-2 nsp3 and MDA5 and prompt to address if IBV nsp3 could also interact with MDA5. As shown in Figure 2b, Hibit-tagged IBV nsp3 could be efficiently co-precipitated with Flag-MDA5 when the two proteins were co-expressed in cells.

The interaction between IBV nsp3 and MDA5 was then studied in virus-infected cells. For this purpose, a recombinant IBV harboring an HA-tagged nsp3 at the nucleotide position 3541 (rIBV-HA-3541) was rescued and used for immunoprecipitation assay with anti-HA coated beads. The H1299 cells infected with wild-type (rIBV-WT) and rIBV-HA-3541 were harvested at 20 h post-infection and analyzed by Western blot with anti-HA antibodies, showing efficient detection of the HA-tagged nsp3 in cells infected with rIBV-HA-3541, but not in cells infected with rIBV-WT (Figure 2c). The endogenous MDA5 was detected in cells infected with either rIBV-WT or rIBV-HA-3541 by Western blot (Figure 2c). However, the immunoprecipitation of cell lysates with anti-HA antibodies detected the endogenous MDA5 only in cells infected with rIBV-HA-3541 (Figure 2c). These results confirm the interaction between nsp3 and MDA5 in IBV-infected cells.

The immunofluorescent staining of the H1299 cells infected with rIBV-HA-3541 was carried out to further study the interaction between MDA5 and IBV nsp3. At 20 h post-infection, the cells were co-stained with anti-HA and anti-MDA5 antibodies and examined. The fluorescent images of MDA5 were detected in the cytoplasm in both infected and uninfected cells, and fluorescent images of the HA-tagged IBV nsp3 were mainly detected in the perinuclear region, the viral replication site, and in the rIBV-HA-3541-infected cells (Figure 2d). Interestingly, in HA-tagged IBV nsp3-positive cells, significantly more MDA5 fluorescence overlapped with the nsp3 images was observed (Figure 2d), suggesting the enrichment of MDA5 in the viral replication site by the interaction between the two proteins. Taken together, these results verified the direct interaction between SARS-CoV-2 nsp3 and MDA5 in the absence of other viral proteins and RNA and further demonstrated the interaction between IBV nsp3 and MDA5 in IBV-infected cells as well as in cells overexpressing the two proteins. This interaction may position MDA5 to the viral RNA synthesis site to modulate the PAMP detection.

### 3.3. Mapping of the SARS-CoV-2 nsp3 Domain(s) Responsible for Its Interaction with MDA5

We next set up to map the domain(s) in SARS-CoV-2 nsp3 responsible for interacting with MDA5. Five deletion constructs, nsp3-N1, nsp3-N2, nsp3-N3, nsp3-M1, and nsp3-C, containing N-terminal amino acids 1–1076, 1–812, 1–386, amino acids 744–1076, and C-terminal amino acids 1077–1944, respectively, were constructed based on SARS-CoV-2 nsp3 (Figure 3a). Co-transfection of MDA5 with each of these constructs showed efficient expression of the Flag-tagged MDA5 and the expected truncated nsp3 products (Figure 3b). Immunoprecipitation with anti-Flag antibodies showed that all five truncated SARS-CoV-2 nsp3 proteins could be co-precipitated with MDA5 (Figure 3b), indicating the presence of more than one domain in the SARS-CoV-2 nsp3 that may interact with MDA5.

### 3.4. Downregulation of the MDA5-Mediated IFN-β Induction by SARS-CoV-2 and IBV nsp3

The functional significance of the interaction between coronavirus nsp3 and MDA5 was then explored by checking the IFN-β production in the cells co-expressing the two proteins. In the cells co-transfected with same amounts of Hibit-SARS2-nsp3 and Flag-MDA5, efficient expression of Flag-MDA5 and Hibit-SARS-2 nsp3 was detected (Figure 4a). The overexpression of Flag-MDA5 induced 20-fold enhanced IFN-β transcription at 24 h post-transfection, compared with the control group transfected with the empty vector (Figure 4b). The IFN-β induction level was reduced by nearly half in cells co-transfected with MDA5 and SARS-CoV-2 nsp3 (Figure 4b). Similar results were also obtained in cells co-expressing Flag-MDA5 and Hibit-IBV-nsp3 (Figure 4a,b). These results demonstrate that the interaction between coronavirus nsp3 and MDA5 suppresses the MDA5-mediated IFN-β induction.

The domain(s) responsible for this suppressive effect was then investigated by co-expression of MDA5 with one of the four truncated SARS-CoV-2 constructs, nsp3-N1, nsp3-N2, nsp3-N3, and nsp3-C (see Figure 3a). The co-transfection of MDA5 with each of these constructs showed efficient expression of MDA5 and the truncated nsp3 products (Figure 4a,c), and the most significant reduction in the IFN-β production was observed in cells co-transfected with nsp3-N1 and MDA5 (Figure 4d). This suppressive effect was almost but not fully relieved by the deletion of the PLpro domain in nsp3-N2 (Figure 4d). Interestingly, significant enhancement of the IFN-β induction was detected in cells overexpressing MDA5 with either nsp3-N3 or nsp3-C (Figure 4d), pointing to the possibility that these two regions may contain sequences that promote the MDA5-mediated IFN-β induction.

### 3.5. Requirement of the Catalytic Activity of PLpro Domain in SARS-CoV-2 and IBV nsp3 for Suppressing the MDA5-Mediated IFN-β Induction

To further support the involvement of the PLpro domain in suppressing the MDA5-mediated IFN-β induction, the catalytic triad of SARS-CoV-2 PLpro (Cys857, His1021, and Asp1032) and IBV PLpro (Cys601, His764, and Asp775) [22,23], were substituted with an Ala, respectively, and efficiently expressed (Figure 5a–c). Compared with wild-type nsp3, overexpression of all mutant constructs failed to suppress the MDA5-mediated IFN-β induction (Figure 5d,e), confirming that the catalytic activity of PLpro is essential for its role in the suppression of the MDA5-mediated IFN-β induction. It was also noted that the co-expression of MDA5 with either SARS-CoV-2 or IBV nsp3 did not result in cleavage of MDA5 in this and previous expression studies (Figure 5b,c), demonstrating that this suppressive effect is not caused by the nsp3-mediated proteolysis of the MDA5 molecule. Instead, the DUB activity of the PLpro domain would be responsible for this effect.

### 3.6. Enhancement of the MDA5-Mediated IFN-β Induction by Ubiquitylation

MDA5 and ubiquitin (Ub) were co-transfected into HEK293T cells to explore the possibility that the de-ubiquitination activity of coronavirus nsp3 may affect the ubiquitination of MDA5 and thus render a suppressive effect on the production of IFN-β. The co-expression of MDA5 with Ub significantly promoted the MDA5-mediated IFN-β induction, and this enhancement effect was suppressed by co-transfection with either SARS-CoV-2 or IBV nsp3 (Figure 6a,b).

This possibility was further studied by transfection of cells with wild-type SARS-CoV-2 and IBV nsp3 as well as mutant nsp3 constructs (SARS2-nsp3-H1021A and IBV-nsp3-H764A). The expression of wild-type nsp3 from either IBV or SARS-CoV-2 significantly reduced the ubiquitination of MDA5 (Figure 6c), but over-expression of the two mutant nsp3 constructs failed to do so. These results confirm that the proteolytic activity of coronavirus nsp3 is essential for its DUB function, and this DUB activity inhibits MDA5 ubiquitination and suppresses the MDA5 signaling.

### 3.7. Interaction of Ubiquitinated Proteins with nsp3 and MDA5

As ubiquitination and de-ubiquitylation of MDA5 and coronavirus nsp3 play essential regulatory roles in the MDA5-mediated innate immune response, we next set up to study if Ub forms complexes with nsp3 and MDA5 and the functional impact of these modifications on the interaction between nsp3 and MDA5. In cells overexpressing Ub, significantly more MDA5 ubiquitination was observed (Figure 7a). Immunoprecipitation with anti-Myc antibodies against the Myc-tagged Ub showed co-precipitation of MDA5 (Figure 7a). In the cells co-transfected with Myc-tagged Ub and HA-tagged SARS-CoV-2 nsp3, considerably more ubiquitinated proteins and MDA5 were co-precipitated with HA-tagged nsp3 by anti-HA antibodies (Figure 7b). Similarly, in cells co-transfected with Myc-tagged Ub and Flag-tagged IBV nsp3, significantly more ubiquitinated proteins and MDA5 were co-precipitated with Flag-tagged IBV nsp3 by anti-Flag antibodies (Figure 7c). These results confirm the formation of complexes among coronavirus nsp3, MDA5 and ubiquitinated proteins.

## 4. Discussion

Coronavirus nsp3 is the largest mature nsp encoded by the genome, with an average molecular mass of about 200 kDa and containing multi-putative functional domains [24]. The multi-functional properties of this protein provide multiple research directions for the identification of novel antiviral targets. However, most current studies were carried out by infection with relevant viruses and by the overexpression of a certain nsp3 domain [24]. In this study, we established a two-plasmid system using K1E phage RNA polymerase and a related powerful promoter to efficiently express nsp3 from SARS-CoV-2 and IBV. Using the expression system, we showed that nsp3 from both IBV and SARS-CoV-2 could physically interact with MDA5 in cells overexpressing nsp3 as well as in IBV-infected cells. This interaction would enrich MDA5 to the viral RNA synthesis site and modulate its PAMP detection, leading to the suppression of the MDA5-mediated IFN-β induction. This suppressive effect was further shown to be associated with the DUB activity of the PLpro domain from both IBV and SARS-CoV-2 nsp3.

Ubiquitination is one of the important post-translational modifications and an important step in the activation of PRRs. Regulating innate immune pathways through antagonizing ubiquitin and ubiquitin-like modifications by viral proteases is recognized as a common mechanism [25]. For example, the delivery of K63-linked polyubiqutin moiety to the CARDs of RIG-I can activate downstream signal transduction, and the K172 ubiquitination is critical for inducing type I IFN expression [26,27,28]. It was also reported that the binding of ISG15 to CARDs of MDA5 promotes the formation of a higher-order MDA5 assembly, triggering the activation of innate immunity against a range of viruses, including coronaviruses, flaviviruses, and picornaviruses [21].

Coronavirus nsp3 is implicated in cleaving proteinaceous post-translational modifications on host proteins as an evasion mechanism against host antiviral immune responses [12,29]. The DUB and deISGylating activities of nsp3 can remove Ub and ISG15 from cellular substrates in addition to cleavage of viral polyproteins to form a functional replicase complex [23]. SARS-CoV-2 nsp3 has been shown to recognize Lys48-linked polyubiquitin via S1 and S2 Ub-binding sites and remove it efficiently [30]. ISG15-dependent MDA5 activation is antagonized by the deISGylation of the SARS-CoV-2 PLpro region [31]. Furthermore, different coronavirus nsp3 may have different preferences for these substrates. For example, SARS-CoV nsp3 may predominantly target Ub chains, but SARS-CoV-2 nsp3 has a preference for cleaving ISG15 [31]. In this study, we show that SARS-CoV-2 and IBV nsp3 suppress the ubiquitination of MDA5 and the MDA5-mediated IFN induction with a similar efficiency.

The functions of other nsp3 domains, such as UBL1/2 and X-domain, in the regulation of IFN signaling are not fully understood. It has been reported that SARS-CoV nsp3 is antagonistic to the expression of type I IFN and does not only depend on the DUB activity [32], as the addition of inhibitors blocking DUB activity cannot completely eliminate this antagonism. This is consistent with our observations that PLpro played the strongest antagonistic role in MDA5-mediated IFN-β induction, but UBL1 and UBL2 also showed a certain inhibitory effect. Ubiquitination is also known to be closely related to NF-κB activation. Polyubiquitination of receptor-interacting protein (RIP), TNF receptor-associated factor 6 (TRAF6), and TNF receptor-associated factor 2 (TRAF2) activates these signaling intermediates, leading to the polyubiquitination of IκB [33]. It is likely that coronavirus nsp3 could not only inhibit MDA5 recognition of viral RNA but also affect the activation of other key factors in the type I IFN pathway. Further studies are required to further clarify these regulatory mechanisms.

The interaction between coronavirus nsp3 and MDA5 would lead to the deubiquitination of MDA5 and inhibition of the MDA5-mediated antiviral signal transduction, probably through interaction with the PLpro domain. In fact, the cotransfection of wild-type and mutant nsp3 PLpro domain from different coronaviruses confirmed the interaction between the PLpro domain and the MDA5-2CARD [21]. In addition to the interaction between the nsp3 PLpro domain and MDA5, at least two other domains, located at the N- and C-terminal regions, respectively, of nsp3, were found to be interacting with MDA5. Interestingly, co-expression of these two regions, each with MDA5, enhanced the MDA5-mediated IFN induction. It is currently unknown if these two regions can directly interact with MDA5 or through interaction with ubiquitination of the proteins.

In addition to evading the type I IFN pathway by nsp3, coronaviruses have evolved a diversity of other counter-measuring mechanisms. For example, Oh et al. demonstrated that SARS-CoV-2 N protein could inhibit the interaction between tripartite motif protein 25 (TRIM25) and RIG-I. N protein can also affect the expression of Tank-binding kinase 1 (TBK1), blocking the nuclear translocation of IRF3 [34]. Wang et al. found that SARS-CoV-2 nsp12 attenuated the IFN-β promoter activity induced by the components of the RIG-I/MDA5 pathway or Sendai virus (SeV) infection [35]. Zhu X has certified that over-expression of porcine deltacoronavirus (PDCoV) nsp5 inhibited the transduction of IFN signal through cleaving STAT2, an important component of the transcription factor complex of ISGF3 [36]. It was also reported that IBV infection disrupts the MDA5 signaling pathway by cleavage of the adaptor protein MAVS, which is essential for the activation of NF-κB and IRF3/7 downstream signaling pathways [37].

In conclusion, this study confirms that SARS-CoV-2 and IBV nsp3 functions as type I IFN antagonist by inhibiting the MDA5-mediated signal transduction pathway through multiple interactions with MDA5. Both nsp3 and MDA5 were shown to interact with Ub, and the DUB activity of nsp3 is essential for its IFN antagonistic function. This information would be of help in designing anti-coronaviral interventions.

## Figures and Tables

**Figure 1 ijms-23-11692-f001:**
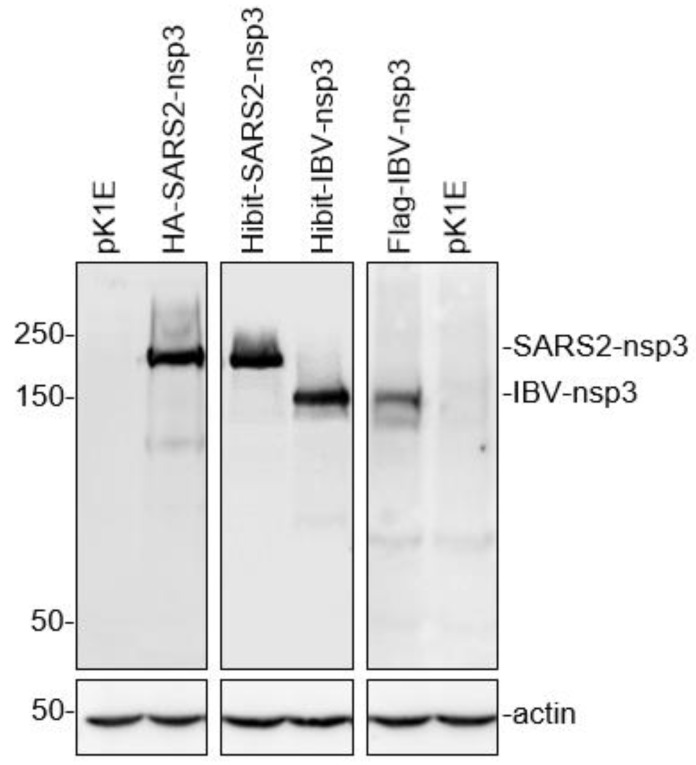
Efficient expression of the full-length nsp3 from SARS-CoV-2 and IBV.

**Figure 2 ijms-23-11692-f002:**
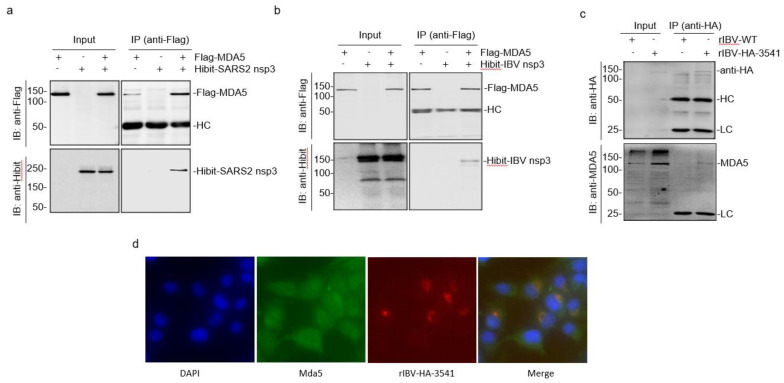
Interaction of MDA5 with SARS-CoV-2 and IBV nsp3. (**a**) Co-immunoprecipitation of SARS-CoV-2 nsp3 with MDA5. HEK293T cells were transfected with Flag-MDA5 and Hibit-tagged SARS-CoV-2 nsp3 (Hibit-SARS2-nsp3), harvested at 24 h post-infection, total cell lysates prepared and subjected to immunoprecipitation with anti-Flag beads. Total cell lysates (Input) and precipitates [IP (anti-Flag)] were analyzed by Western blot with anti-Flag and anti-Hibit antibodies, respectively. Numbers on the left indicate protein sizes in kilodalton. (**b**) Co-immunoprecipitation of IBV nsp3 with MDA5. HEK293T cells were transfected with Flag-MDA5 and Hibit-tagged IBV nsp3 (Hibit-IBV-nsp3), harvested at 24 h post-infection, total cell lysates prepared, and subjected to immunoprecipitation with anti-Flag beads. Total cell lysates and precipitates were analyzed by Western blot with anti-Flag and anti-Hibit antibodies, respectively. Numbers on the left indicate protein sizes in kilodalton. (**c**) Co-immunoprecipitation of IBV nsp3 with MDA5 in IBV-infected cells. Confluent H1299 cells were infected with wild-type (rIBV-WT) and rIBV-HA-3541-nsp3 at an MOI~2, respectively. Cells were harvested at 20 h post-infection, total cell lysates prepared and subjected to immunoprecipitation with anti-HA beads. Total cell lysates and precipitates were analyzed by Western blot with anti-HA and anti-MDA5 antibodies, respectively. Numbers on the left indicate protein sizes in kilodalton. (**d**) Colocalization of the endogenous MDA5 with the HA-tagged IBV nsp3 in H1299 cells infected with rIBV-HA-3541-nsp3. Cells were infected with rIBV-HA-3541-nsp3 at an MOI~2, fixed at 24 h post-infection, permeabilized, and double-labeled with mouse anti-HA and rabbits anti-MDA5 antibodies. MDA5 was immunostained with Alexa Fluor 488-linked anti-rabbit IgG, the HA-nsp3 was stained with Alexa Fluor 594-linked anti-mouse IgG, and cellular nuclei were stained with DAPI (Hoechst 33342).

**Figure 3 ijms-23-11692-f003:**
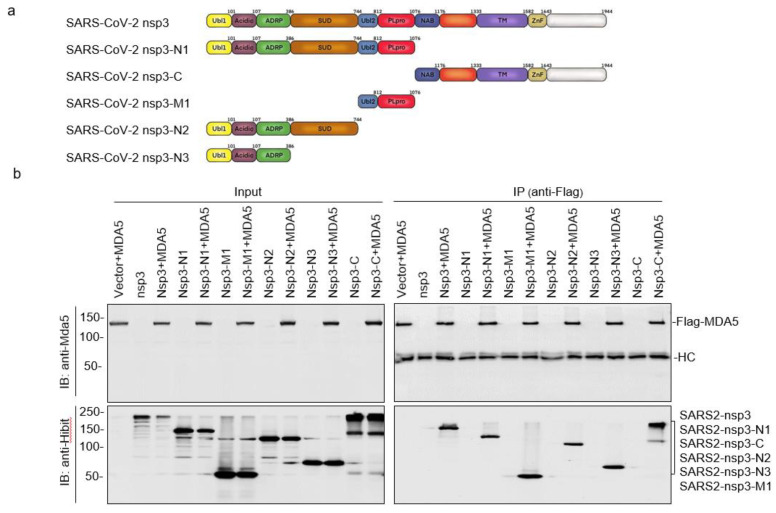
Mapping of the SARS-CoV-2 nsp3 domain(s) responsible for the interaction with MDA5. (**a**) Diagram showing the putative functional domains in the full-length and five truncated SARS-CoV-2 nsp3 constructs, nsp3-N1, nsp3-C, nsp3-M1, nsp3-N2, and nsp3-N3. (**b**) Co-immunoprecipitation of MDA5 with wild-type and truncated SARS-CoV-2 nsp3 constructs. HEK293T cells were transfected with wild-type and truncated SARS-CoV-2 nsp3 constructs, harvested at 24 h post-transfection, lysed with RIPA buffer, and subjected to immunoprecipitation with anti-Flag beads. Total cell lysates and precipitates were analyzed by Western blot with anti-Flag and anti-Hibit antibodies, respectively. Numbers on the left indicate protein sizes in kilodalton.

**Figure 4 ijms-23-11692-f004:**
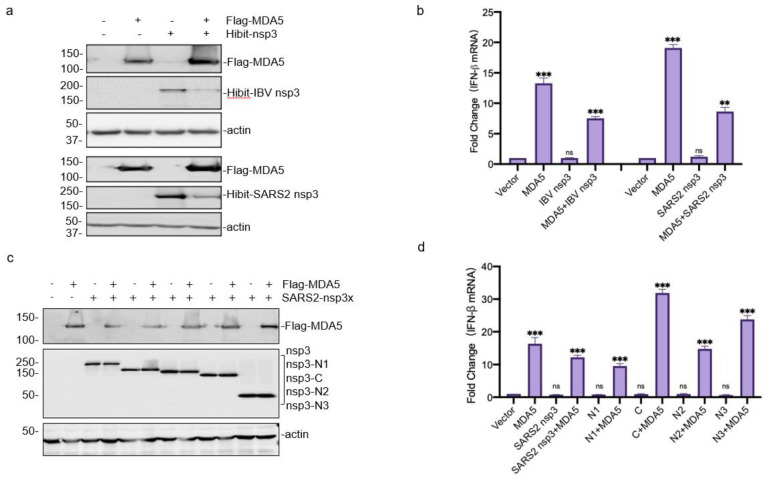
Suppression of the MDA5-mediated IFN-β induction by SARS-CoV-2 and IBV nsp3. (**a**) Western blot analysis of the expression of Flag-MDA5 co-transfected with Hibit-tagged SARS-CoV-2 or IBV nsp3. HEK293T cells were transfected either with Flag-MDA5 alone or together with Hibit-tagged SARS-CoV-2 or IBV nsp3. Cells were harvested at 24 h post-transfection and subjected to Western blot analysis with anti-FLAG and anti-Hibit antibodies, respectively. Beta-actin was included as the loading control. Sizes of protein ladders in kDa are indicated on the left. (**b**) Suppression of the MDA5-mediated IFN-β induction by SARS-CoV-2 and IBV nsp3. Total RNA was extracted from cells harvested in (**a**), and the IFN-β levels were determined by RT-qPCR after normalization with the GAPDH mRNA extracted from cells transfected with the empty vector. Significance levels were presented by the *p*-value (ns, non-significance, **, *p* < 0.01; ***, *p* < 0.001). (**c**) Western blot analysis of the expression of Flag-MDA5 co-transfected with Hibit-tagged full-length or truncated SARS-CoV-2 nsp3. Cells were harvested at 24 h post-transfection and subjected to Western blot analysis with anti-FLAG and anti-Hibit antibodies, respectively. Beta-actin was included as the loading control. Sizes of protein ladders in kDa are indicated on the left. (**d**) Effects of co-expression of MDA5 and the full-length or truncated SARS-CoV-2 nsp3 on the MDA5-mediated IFN-β induction. Total RNA was extracted from cells harvested in (**c**), and the IFN-β levels were determined by RT-qPCR after normalization with the GAPDH mRNA extracted from cells transfected with the empty vector. Significance levels were presented by the *p*-value (ns, non-significance, ***, *p* < 0.001).

**Figure 5 ijms-23-11692-f005:**
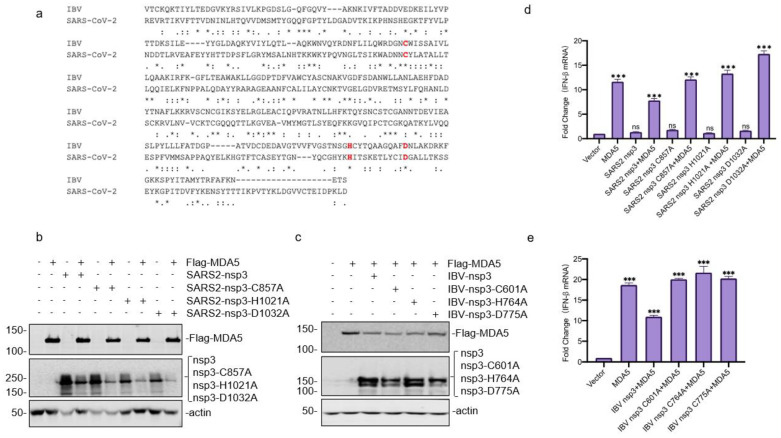
Requirement of the catalytic activity of the PLpro domain for suppressing the MDA5-mediated IFN-β induction. (**a**) Alignment of the PLpro regions from IBV and SARS-CoV-2 nsp3. The alignment was produced using Mafft and sequences obtained from NCBI. The catalytic triad, as well as residues mutated in this study, are indicated in red. (**b**) Western blot analysis of the expression of Flag-MDA5 co-transfected with Hibit-tagged wild-type and mutant SARS-CoV-2 nsp3. HEK293T cells were transfected either with Flag-MDA5 alone or together with Hibit-tagged wild-type or mutant SARS-CoV-2 nsp3. Cells were harvested at 24 h post-transfection and subjected to Western blot analysis with anti-FLAG and anti-Hibit antibodies, respectively. Beta-actin was included as the loading control. Sizes of protein ladders in kDa are indicated on the left. (**c**) Western blot analysis of the expression of Flag-MDA5 co-transfected with Hibit-tagged wild-type and mutant IBV nsp3. HEK293T cells were transfected either with Flag-MDA5 alone or together with Hibit-tagged wild-type or mutant IBV nsp3. Cells were harvested at 24 h post-transfection and subjected to Western blot analysis with anti-FLAG and anti-Hibit antibodies, respectively. Beta-actin was included as the loading control. Sizes of protein ladders in kDa are indicated on the left. (**d**) Effects of co-expression of MDA5 and wild-type or mutant SARS-CoV-2 nsp3 on the MDA5-mediated IFN-β induction. Total RNA was extracted from cells harvested in (**b**), and IFN-β levels were determined by RT-qPCR after normalization with the GAPDH mRNA extracted from cells transfected with the empty vector. Significance levels were presented by the *p*-value (ns, non-significance, *, *p* < 0.05; **, *p* < 0.01; ***, *p* < 0.001). € Effects of co-expression of MDA5 and wild-type or mutant IBV nsp3 on the MDA5-mediated IFN-β induction. Total RNA was extracted from cells harvested in (**e**), and IFN-β levels were determined by RT-qPCR after normalization with the GAPDH mRNA extracted from cells transfected with the empty vector. Significance levels were presented by the *p*-value (ns, non-significance, *, *p* < 0.05; **, *p* < 0.01; ***, *p* < 0.001).

**Figure 6 ijms-23-11692-f006:**
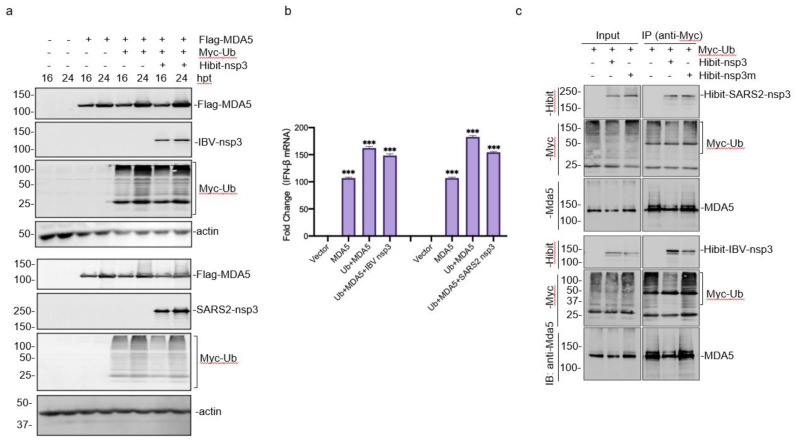
Enhancement of the MDA5-mediated IFN-β induction by ubiquitylation. (**a**) Western blot analysis of the expression of Flag-MDA5 co-transfected with Myc-Ub and Hibit-tagged SARS-CoV-2 or IBV nsp3. HEK293T cells were transfected either with Flag-MDA5 alone or together with Myc-Ub, and Hibit-tagged SARS-CoV-2 or IBV nsp3. Cells were harvested at 16 and 24 h post-transfection (hpt), respectively, and subjected to Western blot analysis with anti-FLAG and anti-Hibit antibodies, respectively. Beta-actin was included as the loading control. Sizes of protein ladders in kDa are indicated on the left. (**b**) Effects of co-expression of MDA5 with Ub and SARS-CoV-2 or IBV nsp3 on the MDA5-mediated IFN-β induction. Total RNA was extracted from cells harvested in (**a**), and IFN-β levels were determined by RT-qPCR after normalization with the GAPDH mRNA extracted from cells transfected with the empty vector. Significance levels were presented by the *p*-value (ns, non-significance, ***, *p* < 0.001). (**c**) Co-immunoprecipitation of Ub with MDA5 and wild-type and mutant SARS-CoV-2 or IBV nsp3. HEK293T cells were transfected with Myc-Ub alone or together with Hibit-tagged wild-type and mutant SARS-CoV-2 or IBV nsp3, harvested at 24 h post-infection, total cell lysates prepared, and subjected to immunoprecipitation with anti-Myc beads. Total cell lysates and precipitates were analyzed by Western blot with anti-Myc, anti-MDA5, and anti-Hibit antibodies, respectively. Numbers on the left indicate protein sizes in kilodalton.

**Figure 7 ijms-23-11692-f007:**
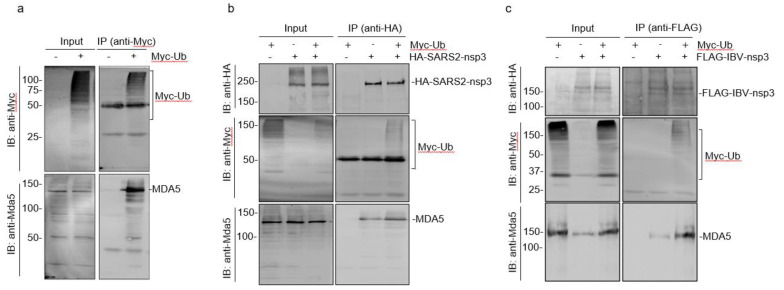
Interaction of ubiquitin and ubiquitinated proteins with nsp3 and MDA5. (**a**) Co-immunoprecipitation of Ub with MDA5. HEK293T cells were transfected with Myc-Ub, harvested at 24 h post-infection, total cell lysates prepared, and subjected to immunoprecipitation with anti-Myc beads. Total cell lysates and precipitates were analyzed by Western blot with anti-Myc and anti-MDA5 antibodies, respectively. Numbers on the left indicate protein sizes in kilodalton. (**b**) Co-immunoprecipitation of HA-tagged SARS-CoV-2 nsp3 with Myc-Ub and MDA5. HEK293T cells were transfected with Myc-Ub alone or together with HA-tagged SARS-CoV-2 nsp3, harvested at 24 h post-infection, total cell lysates prepared, and subjected to immunoprecipitation with anti-HA beads. Total cell lysates and precipitates were analyzed by Western blot with anti-Myc, anti-MDA5, and anti-HA antibodies, respectively. Numbers on the left indicate protein sizes in kilodalton. (**c**) Co-immunoprecipitation of Flag-tagged IBV nsp3 with Myc-Ub and MDA5. HEK293T cells were transfected with Myc-Ub alone or together with Flag-tagged IBV nsp3, harvested at 24 h post-infection, total cell lysates prepared, and subjected to immunoprecipitation with anti-Flag beads. Total cell lysates and precipitates were analyzed by Western blot with anti-Myc, anti-MDA5 and anti-Flag antibodies, respectively. Numbers on the left indicate protein sizes in kilodalton.

## Data Availability

The raw data supporting the conclusion of this article will be made available by the authors, without undue reservation.

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
