# Peer review of "Direct Interaction of Coronavirus Nonstructural Protein 3 with Melanoma Differentiation-Associated Gene 5 Modulates Type I Interferon Response during Coronavirus Infection"

_ijms, 2022, doi:10.3390/ijms231911692_

Round 1

Reviewer 1 Report

1- Not sure why the manuscript is without line numbers

2- Parentheses. Sentence #2 page 3.

3- Introduction. Why include dsRNA while your work’s focus is on ssRNA? page 4.

4-  English. Page 4

When MDA5 recognizes long double-strand RNA, protein phosphatase 1(pp1)α/γ dephosphorylates the CARDs[12], which promotes the binding and activation of mitochondrial antiviral-signaling protein (MAVS), leading to the phosphorylation IRF3 and IRF7, and the induction of type I interferons [13,14].

5- What was the MOI? need to be precise. Page 5

incubated in serum-free DMEM at an MOI of approximately 2.

6- page 5.

since you used TRIzol, did your protocol include DAN cleaning?

7- Did you run RNA quality? 

8- Rewrite. page 6.

The PCR product was inserted into vector pXJ40-Flag by homologous recombination. Plasmid pK1E-Hibit-SARS2-Nsp3 was synthesized by GENEWIZ company, based on the sequence of SARS-CoV-2 nsp3 acquired from NCBI. IBV nsp3 was amplified from total RNA of H1299 cells infected with IBV, and was cloned into pXJ40 and pK1E vectors, respectively, by homologous recombination. All plasmids were verified by sequencing.

9- Add more details—page 7.

After centrifugation and protein concentration determination by spectrophotometer.

10- What was the exact MOI? page 8.

11- Again add more details. For how long?

cells were fixed with ice-cold 100% methanol.

12- Why this was added? clear it for readers.

cells were incubated with 10 µg/mL Hoechst 33342 at room temperature for 5-10 minutes.

13- Could not find (Figure 4d).

14- Needs to be rewritten. 

The possibility that the de-ubiquitination activity of coronavirus nsp3 may affect the ubiquitination of MDA5 and thus render a suppressive effect on the production of IFN-β was explored by co-transfection of cells with MDA5 and ubiquitin (Ub). 

Author Response

Thank you all  for your constructive comments. Based on the comments, we have revised the manuscript as follows. Please see the attachment .

Reviewer 2 Report

The manuscript ijms-1916058 by Sun et al. analyzes the interactions of MDA5 and nsp3 of SARS-CoV-2 and IBV as well as consequences of those interactions with respect of type I IFN production. The authors provided valuable data that support their conclusions. Some minor changes have to be made prior publication.

I suggest the following corrections:

1. Material and Methods section

- all centrifugation speeds have to be indicated as relative centrifugal force (g force)

- Manufacturers of chemicals have to be provided, especially in cases where working dilution (not exact concentration) is indicated

- number and measure units have to be separated by space

- page 6: NCBI accession no. of the sequence of SARS-CoV-2 nsp3 has to be provided

- page 7: the unit for pore size of nitrocellulose membrane has to be corrected (micrometer)

- page 7, the last row: “protease inhibitors” instead of “protein inhibitor”

2. Results

- page 11: “… indicating the presence of at least three domains in the SARS-CoV-2 nsp3 that interact with MDA5” has to be corrected as presented results imply only on the presence of more than one (but not “at least three”) interacting domains.

- page 12, “...The domain(s) responsible for this suppressive effect was then investigated ...”: Why nsp3-M1 was excluded from the analysis?

- The molecular weights has to be indicated on the figures (as it is described in the legend of the figures)

3. Discussion

- Are there some literature data that could explain the opposite effects of truncated nsp3 variants on MDA5-mediated IFNβ production? (reduction in the presence of nsp3-N1 vs. enhancement in the presence of nsp3-N3 and nsp3-C)

Author Response

Thank you for your constructive comments. Based on the comments, we have revised the manuscript as follows. please see the attachment.

Reviewer 3 Report

This paper is very well designed, results have high scientific merit, and presented in adequate form.

Author Response

Thank you for your constructive comments. Based on the comments, we have revised the manuscript as follows. Please see the attachment.
